# Hospital segregation, critical care strain, and inpatient mortality during the COVID-19 pandemic in New York City

**Anna Zhilkova, Laila Alsabahi, Donald Olson, Duncan Maru, Tsu-Yu Tsao[‡]\*, Michelle E. Morse[‡]**

Center for Health Equity and Community Wellness at the New York City Department of Health and Mental Hygiene, Long Island City, NY, United States of America

‡ TYT and MEM are contributed equally to this work as co-senior authors.
\* ttsao@health.nyc.gov

**Data Availability Statement:** This study was produced from data provided by the New York State Department of Health (NYSDOH), and the

## Abstract

### Background

Hospital segregation by race, ethnicity, and health insurance coverage is prevalent, with some hospitals providing a disproportionate share of undercompensated care. We assessed whether New York City (NYC) hospitals serving a higher proportion of Medicaid and uninsured patients pre-pandemic experienced greater critical care strain during the first wave of the COVID-19 pandemic, and whether this greater strain was associated with higher rates of in-hospital mortality.

### Methods

In a retrospective analysis of all-payer NYC hospital discharge data, we examined changes in admissions, stratified by use of intensive care unit (ICU), from the baseline period in early 2020 to the first COVID-19 wave across hospital quartiles (265,329 admissions), and crude and risk-adjusted inpatient mortality rates, also stratified by ICU use, in the first COVID wave across hospital quartiles (23,032 inpatient deaths), based on the proportion of Medicaid or uninsured admissions from 2017–2019 (quartile 1 lowest to 4 highest). Logistic regressions were used to assess the cross-sectional association between ICU strain, defined as ICU volume in excess of the baseline average, and patient-level mortality.

### Results

ICU admissions in the first COVID-19 wave were 84%, 97%, 108%, and 123% of the baseline levels by hospital quartile 1–4, respectively. The risk-adjusted mortality rates for ICU admissions were 36.4 (CI = 34.7,38.2), 43.6 (CI = 41.5,45.8), 45.9 (CI = 43.8,48.1), and 45.7 (CI = 43.6,48.0) per 100 admissions, and those for non-ICU admissions were 8.6 (CI = 8.3,9.0), 10.9 (CI = 10.6,11.3), 12.6 (CI = 12.1,13.0), and 12.1 (CI = 11.6,12.7) per 100 admissions by hospital quartile 1–4, respectively. Compared with the reference group of 100% or less of the baseline weekly average, ICU admissions on a day for which the ICU volume was 101–150%, 151–200%, and > 200% of the baseline weekly average had odds

authors are not authorized by NYSDOH to share the data. If interested, researchers may make a direct request to NYSDOH for access to the data (SPARCS). Additionally, much of SPARCS are publicly available from the link: https://health.data. ny.gov/browse?q=sparcs&sortBy=newest&utf8=% E2%9C%93.

**Funding:** The author(s) received no specific funding for this work.

**Competing interests:** No authors have competing interests.

ratios of 1.17 (95% CI = 1.10, 1.26), 2.63 (95% CI = 2.31, 3.00), and 3.26 (95% CI = 2.82, 3.78) for inpatient mortality, and non-ICU admissions on a day for which the ICU volume was 101–150%, 151–200%, and > 200% of the baseline weekly average had odds ratios of 1.28 (95% CI = 1.22, 1.34), 2.60 (95% CI = 2.40, 2.82), and 3.44 (95% CI = 3.11, 3.63) for inpatient mortality.

## Conclusions

Our findings are consistent with hospital segregation as a potential driver of COVID-related mortality inequities and highlight the need to desegregate health care to address structural racism, advance health equity, and improve pandemic resiliency.

## Introduction

The COVID-19 pandemic has amplified long-standing health inequities in the US [1–3]. In New York City (NYC), the epicenter in the early phase of the pandemic, the inequitable impact of COVID-19 was particularly salient, as persons and communities who historically bear the greatest harms of structural racism and those suffering higher rates of poverty experienced disproportionate rates of COVID-19 infection, hospitalization, and mortality [4–6].

Numerous studies have examined factors related to the inequitable impact of COVID-19 [7–11]. One potential, and under-studied, phenomenon is de facto hospital segregation (henceforth "hospital segregation"), referring to persistence of segregation by health insurance coverage, race, ethnicity, language, and immigration status not sanctioned by law. Hospital segregation has been well documented in NYC and has, among other consequences, contributed to wide inequities in resources for "safety net hospitals", typically referring to hospitals that provide significant proportions of care for underinsured and other vulnerable patients [12–15]. For examples, studies have found that public and other safety-net hospitals provided for a disproportionate share of Medicaid and uninsured care while private academic medical centers (AMCs), generally acknowledged as the most well-resourced hospitals in NYC, were much less likely to admit Black and uninsured patients [13, 15].

With people of color and Medicaid-insured and uninsured individuals disproportionately served by the safety-net system, a greater likelihood of exposure to and development of severe illness from COVID-19 infections among these populations may have led to a surge in the volume of patients needing critical care and straining of the intensive care unit (ICU) capacity among safety-net hospitals [16–20]. While some safety-net hospitals were able to increase ICU-capable beds as a response to surges in ICU demand, it was considerably more challenging to provide and train sufficient staff for ICU care [20]. Moreover, hospitals, especially those that were not part of a larger system, faced additional constraints in their ability to load balance to other hospitals, leaving patients to receive care in overloaded ICUs or experience delays in admission [16, 21] With safety-net providers being chronically underresourced [22, 23], this additional strain on their already stretched-thin capacity may have resulted in higher in-hospital mortality [17].

Studies of hospital outcomes in NYC during the COVID-19 pandemic are mostly based on data or reports from individual hospitals or networks [16–21, 24]. Here, we used all-payer NYC hospital discharge data to provide a population-level analysis across all hospitals to gain a full picture of the crisis citywide and to assess disparities in outcomes in a segregated system of care. Specifically, we examined whether hospitals serving a higher proportion of Medicaid and

uninsured patients pre-COVID pandemic experienced larger increases in inpatient use and greater rates of in-hospital mortality, stratified by ICU use, during the first wave of the COVID-19 pandemic than hospitals serving a lower proportion of Medicaid and uninsured patients pre-COVID pandemic. Additionally, we developed measures of ICU strain and formally assessed whether any excess in-hospital mortality risk was associated with strained ICUs [25, 26]. Due to care segregation and a greater likelihood of COVID-19 infection and hospitalization among Medicaid-insured and uninsured individuals, we hypothesized that (1) hospitals serving a higher proportion of Medicaid and uninsured patients pre-COVID experienced a greater surge of patients requiring critical care and higher mortality rates during the first COVID-19 wave, and (2) the greater surge in critical care patients was positively associated with patient mortality.

## Methods

This retrospective study was approved by the NYC Department of Health and Mental Hygiene (DOHMH) institutional review board as non-human subject research.

### Data and sample

Data on hospital outcomes in NYC were obtained from the Statewide Planning and Research Cooperative System (SPARCS), a comprehensive all-payer data reporting system that collects information on discharges from all Article 28 hospitals in New York State (federal, psychiatric, and long-term care hospitals are excluded). Each discharge record in SPARCS includes information on patient demographics, diagnoses, procedures, sources of payment, charges, and provider and hospital characteristics. Inpatient records from NYC hospitals among individuals aged 18 and older were extracted for all of 2017–2019 (N = 2,435,063) and for January 5-May 9 in 2020 (N = 265,329). Geocoded 2020 SPARCS records were linked to publicly available NYC nursing home addresses using the building identification number to determine if a discharge was attributable to a nursing home resident.

The dates for accessing data used in this analysis spanned from November 2022 to June 2023. The authors had access to information in SPARCS that could identify individual records.

### Measures

**Hospital segregation.** We used 2017–2019 SPARCS data to characterize the extent of pre-existing hospital segregation by health insurance coverage, which correlates highly with race/ethnicity. We ranked hospitals by the proportion of all-cause adult hospitalizations for which the primary payer was Medicaid or self-pay in 2017–2019 and grouped these hospitals into quartiles, with quartile 1 (Q1) including hospitals with the lowest proportions of Medicaid or self-pay admissions, and quartile 4 (Q4) hospitals with the highest proportions of Medicaid or self-pay admissions. Along with Medicaid insured admissions, we focused on self-pay admissions since self-pay patients as a group had the highest concentration of care received from public and private independent hospitals (> 65% in 2017–2019 SPARCS, data not shown), suggesting limited access to well-resourced facilities among this population. Hospitals with a high proportion of Medicaid or self-pay admissions may be referred to as safety nets in the study [27].

**Hospitalizations.** We assessed three sets of hospitalizations: all-cause hospitalizations, ICU hospitalizations, and non-ICU hospitalizations. All-cause hospitalizations included all admissions to NYC hospitals among adults ages 18 and older. ICU admissions were identified based on a validated measure using revenue center codes (RCCs) [28]. Non-ICU admissions

were defined as admissions without the RCCs for ICU use and may include admissions with intermediate ICU use.

**Inpatient mortality.**   We also assessed three sets of in-hospital deaths: deaths among all-cause hospitalizations, deaths among ICU hospitalizations, and deaths among non-ICU hospitalizations. We identified in-hospital death as disposition on discharge status of expired or discharged to hospice. Both crude and risk-adjusted rates of mortality were calculated for hospitals as a group in each quartile. Patient-level characteristics, including sex, age, race/ethnicity, primary payer, comorbidity, admission from nursing home, emergency admission, severity of illness, and principal diagnosis, were used in non-hierarchical logistic regression models to derive predicted probabilities of in-patient death, which were incorporated in the risk-adjusted mortality rate calculations. The clustered sandwich estimator was used to account for potentially correlated outcomes among admissions to the same hospital. After accounting for patient-level factors, any differences in (risk-adjusted) mortality rates across hospitals may be attributable to hospital-level and other factors [29].

**Hospital strain.**   The primary measure of critical care strain on a given day was the total number of ICU admissions to the hospital in the week leading up to that day as a percentage of the hospital's pre-pandemic baseline (defined below) weekly average ICU admissions. This percentage was expressed as one of the following categories: 0–100%, 101–150%, 151–200%, and > 200%.

## Statistical analysis

We performed the following to test the hypothesis that hospitals serving a higher proportion of Medicaid and uninsured patients pre-COVID experienced a greater surge of patients during the first COVID-19 wave. First, we calculated the ratio of hospitalizations for each quartile of hospitals between the two periods in 2020: (1) the pre-pandemic baseline period spanning the dates of January 5-March 7 (CDC weeks 2–10 of 2020), and (2) COVID-19 Wave 1 spanning the dates of March 8-May 9 (CDC weeks 11–19 of 2020), with the ratio = the number of hospitalizations in Wave 1/the number of hospitalization in the baseline. We focused on the first COVID-19 wave and a preceding period of similar length since reports and studies suggest that hospital capacity in NYC was particularly strained in the first wave of the pandemic due to a surge of critically ill patients [16–18]. The standard errors and confidence intervals for the ratios of hospitalizations between the two periods were approximated using Fieller's Theorem [30]. Second, we used t-tests to assess whether the ratios of hospitalizations from higher hospital quartiles were larger than the ratios from lower hospital quartiles ($p < 0.05$).

To test the hypothesis that hospitals serving a larger proportion of Medicaid and uninsured patients pre-COVID experienced higher mortality rates during the first COVID-19 wave, we used t-tests to assess whether crude and risk-adjusted mortality rates for higher quartile hospitals were larger than those for lower quartile hospitals during the first COVID-19 wave ($p < 0.05$). Between these two measures of mortality, risk-adjusted rates may be of greater relevance since it is necessary to account for differences in patient characteristics across hospitals in assessing the possibility of heightened mortality rates resulting from hospital-level factors such as constrained capacity.

In addition to performing statistical testing of differences across hospital quartiles for the entire first wave of COVID-19 and the preceding baseline, we calculated weekly admissions and weekly in-hospital deaths as percentages of baseline weekly averages and illustrated them graphically to discern the weeks in which there may be particularly large disparities in outcomes. We also provided information on the characteristics of admissions and deaths from the baseline and Wave 1 for all hospital quartiles to better understand distributions of COVID-19 related burdens among various demographic groups.

We used both the hierarchical logistic regression model with hospital random intercept and non-hierarchical logistic regression model with hospital fixed effects to assess the association between hospital strain and patient-level mortality in the ICU setting, in the non-ICU setting, and for all adult admissions. In addition to ICU strain, the following patient-, neighborhood-, and hospital-level covariates were included in the model: sex (female), age (18–39, 40–64, 65–79, 80 or older), race/ethnicity (Asian or Pacific Islander, Black, Hispanic, White, and Other non-Hispanic), primary payer (Medicaid, Medicare, private/commercial, self-pay, and Other), weighted Elixhauser comorbidity index, admission from nursing home, measures of patient acuity, including emergency admission and severity of illness (SOI) (SOI present on admission is based on All Patient Refined Diagnosis Related Groups (APR-DRG) classification and coded as 1 = minor, 2 = moderate, 3 = major, and 4 = extreme), and principal diagnosis based on Clinical Classification Software Refined (CCSR) version 2021.1 (including COVID-19 identification based on ICD-10 code U.071 in effect April 1, 2020), neighborhood of residence (defined as Neighborhood Tabulation Area), and hospital type (private AMC and affiliates, private other, and public).

Several alternative proxies for hospital strain were used to assess sensitivity of regression estimates to hospital strain measures, including (1) daily average adult admissions with ICU use over a 7-day period leading to the date of admission as a percentage of certified ICU beds, (2) adult COVID-19 positive admissions with ICU use over a 7-day period leading to the date of admission as a percentage of adult admissions with ICU use, and (3) adult sepsis admissions over a 7-day period leading to the date of admission in any setting as a percentage of pre-pandemic weekly average adult sepsis admissions.

Data preparation and analyses were performed in R version 4.2.3 and Stata version 16, respectively.

## Results

### Pre-existing segregation of care (2017–2019)

Table 1 shows the characteristics of admissions and hospitals across the four quartiles of hospitals in 2017–2019. After excluding specialty hospitals and hospitals without ICUs, a total of 47 hospitals in three categories–private AMC and affiliates, private other, and public–were included for the study. Q1 hospitals, almost all of which were AMCs or AMC affiliates, accounted for the highest proportion of all admissions in NYC at 33% (802,657/2,437,334). The proportion of admissions provided by each quartile steadily declined to 19% (464,936/2,437,334) among Q4 hospitals, all of which were public or private independent. Almost all public hospitals in NYC are part of the H+H system, which is run by the city and a primary provider of care for Medicaid and uninsured patients. The proportion of admissions for which the primary payer was Medicaid or self-pay varied substantially, from 22% in Q1 to 61% in Q4, confirming a high degree of hospital segregation by insurance coverage in NYC. Provision of care for Medicaid or self-pay admissions disproportionately fell on public and non-AMC affiliated private hospitals (hospitals in Q3 and Q4). In addition to substantially higher proportions of Medicaid or self-pay admissions, hospitals in the higher quartiles tended to have admissions through emergency department (ED) visits and admissions associated with younger patients, Black or Hispanic patients, patients from higher poverty neighborhoods, and patients who were NYC residents.

### Admissions and in-hospital mortality (2020)

Table 2 shows the numbers of admissions in the pre-pandemic baseline period and during COVID-19 Wave 1 and the ratios of admissions between these two periods across the four

**Table 1. Characteristics of hospitalization admissions among adults ages 18+ in NYC by hospital quartile, 2017–2019[a].**

| Characteristic, No. (%) | Overall | Quartile 1 | Quartile 2 | Quartile 3 | Quartile 4 |
|---|---|---|---|---|---|
| Hospital Type[b] | (N = 47) | (N = 11) | (N = 12) | (N = 12) | (N = 12) |
| Private AMC | 22 | 10 | 9 | 3 | 0 |
| Private other (non-AMC) | 13 | 1 | 2 | 8 | 2 |
| Public | 12 | 0 | 1 | 1 | 10 |
| All admissions | 2,437,334 (100%) | 802,657 (100%) | 655,001 (100%) | 514,740 (100%) | 464,936 (100%) |
| Admissions through emergency department | 1,646,166 (68%) | 465,637 (58%) | 439,671 (67%) | 378,413 (74%) | 362,445 (78%) |
| Gender | | | | | |
| Female | 1,384,740 (57%) | 477,796 (60%) | 384,248 (59%) | 288,076 (56%) | 234,620 (50%) |
| Male | 1,052,536 (43%) | 324,853 (40%) | 270,738 (41%) | 226,651 (44%) | 230,294 (50%) |
| Age group | | | | | |
| Age (median) | 57 | 59 | 61 | 56 | 51 |
| Age 18–39 | 681,563 (28%) | 220,988 (28%) | 155,927 (24%) | 152,526 (30%) | 152,112 (33%) |
| Age 40–64 | 833,922 (34%) | 249,867 (31%) | 217,679 (33%) | 172,550 (34%) | 193,826 (42%) |
| Age 65–79 | 560,271(23%) | 203,326 (25%) | 167,785 (26%) | 108,051 (21%) | 81,109 (17%) |
| Age 80+ | 357,707 (15%) | 127,179 (16%) | 112,398 (17%) | 80,624 (16%) | 37,506 (8.1%) |
| Race/ethnicity | | | | | |
| NH Asian or Pacific Islander | 200,814 (8.2%) | 49,562 (6.2%) | 84,037 (13%) | 47,755 (9.3%) | 19,460 (4.2%) |
| NH Black | 618,876 (25%) | 142,739 (18%) | 164,967 (25%) | 130,655 (25%) | 180,515 (39%) |
| Hispanic | 539,405 (22%) | 121,707 (15%) | 154,824 (24%) | 105,772 (21%) | 157,102 (34%) |
| NH White | 723,216 (30%) | 369,061 (46%) | 171,911 (26%) | 146,354 (28%) | 35,890 (7.7%) |
| Other | 355,023 (15%) | 119,588 (15%) | 79,262 (12%) | 84,204 (16%) | 71,969 (15%) |
| Insurance | | | | | |
| Medicaid | 853,173 (35%) | 167,707 (21%) | 200,253 (31%) | 224,589(44%) | 260,624 (56%) |
| Medicare | 1,012,871 (42%) | 360,525 (45%) | 307,057 (47%) | 212,348 (41%) | 132,941 (29%) |
| Private | 501,370 (21%) | 254,335 (32%) | 137,517 (21%) | 65,679 (13%) | 43,539 (9.4%) |
| Self-pay | 45,586 (1.9%) | 9,074 (1.1%) | 6,649 (1.0%) | 7,901 (1.5%) | 21,962 (4.7%) |
| Other | 24,334 (1%) | 11,016 (1.4%) | 3,525 (0.5%) | 3,923 (0.8%) | 5,870 (1.3%) |
| Neighborhood poverty[c] | | | | | |
| 0 to <10% | 394,154 (16%) | 184,361 (23%) | 130,829 (20%) | 43,202 (8.4%) | 35,762 (7.7%) |
| 10 to <20% | 930,296 (38%) | 270,931 (34%) | 268,531 (41%) | 231,309 (45%) | 159,525 (34%) |
| 20 to <30% | 498,747 (20%) | 145,737 (18%) | 107,272 (16%) | 148,191 (29%) | 97,547 (21%) |
| 30% or higher | 391,179 (16%) | 71,619 (8.9%) | 82,767 (13%) | 76,428 (15%) | 160,365 (34%) |
| Borough | | | | | |
| Bronx | 493,240 (20%) | 56,305 (7.0%) | 200,668 (31%) | 57,497 (11%) | 178,770 (38%) |
| Brooklyn | 667,904 (27%) | 225,457 (28%) | 59,527 (9.1%) | 274,190 (53%) | 108,730 (23%) |
| Manhattan | 379,641 (16%) | 231,607 (29%) | 40,042 (6.1%) | 42,357 (8.2%) | 65,635 (14%) |
| Queens | 506,025 (21%) | 71,840 (9%) | 226,683 (35%) | 109,277 (21%) | 98,225(21%) |
| Staten Island | 150,531 (6.2%) | 85,730 (11%) | 57,258 (8.7%) | 6,166 (1.2%) | 1,377 (0.3%) |
| NYC Residency | | | | | |
| NYC residents | 2,197,341 (90%) | 670,939 (84%) | 584,178 (89%) | 489,487 (95%) | 452,737(97%) |
| Non-NYC residents | 239,993 (10%) | 131,718 (16%) | 70,823 (11%) | 25,253 (4.9%) | 12,189 (2.6%) |

Notes

Abbreviations: NYC = New York City; NH = non-Hispanic; AMC = academic medical center (including affiliates).

[a] Hospital quartiles are defined based on a hospital's percentage of adult hospitalizations where the primary payer was Medicaid or self-pay. Hierarchical algorithm based on common payment logic was used to identify primary payer.

[b] Excludes specialty hospitals and hospitals without intensive care units (N = 6).

[c] Poverty estimates may not be available for all NYC neighborhoods.

Source: New York State Department of Health, Statewide Planning and Research Cooperative System (SPARCS) 2017–2019 inpatient data, 2022 release.

**Table 2. Admissions and ratios of admissions among adults ages 18+ in NYC by hospital quartile, January 5-May 9, 2020[a].**

|  | NYC | | Quartile 1 (lowest % of Medicaid or Self-pay patients) | | Quartile 2 | | Quartile 3 | | Quartile 4 (highest % of Medicaid or Self-pay patients) | |
|---|---|---|---|---|---|---|---|---|---|---|
|  | (N = 47 hospitals)[b] | | (N = 11 hospitals) | | (N = 12 hospitals) | | (N = 12 hospitals) | | (N = 12 hospitals) | |
| **Admission type** | Pre-pandemic[c] | Wave 1[d] | Pre-pandemic | Wave 1 | Pre-pandemic | Wave 1 | Pre-pandemic | Wave 1 | Pre-pandemic | Wave 1 |
| All admissions | 145,054 (100%) | 120,275 (100%) | 49,977 (100%) | 36,257 (100%) | 38,360 (100%) | 32,062 (100%) | 29,846 (100%) | 26,020 (100%) | 26,871 (100%) | 25,936 (100%) |
| ICU admissions[e] | 16,108 (11%) | 15,948 (13%) | 5,986 (12%) | 5,015 (14%) | 3,799 (10%) | 3,680 (11%) | 3,484 (12%) | 3,770 (14%) | 2,839 (11%) | 3,483 (13%) |
| Non-ICU admissions[f] | 128,946 (89%) | 104,327 (87%) | 43,991 (88%) | 31,242 (86%) | 34,561 (90%) | 28,382 (89%) | 26,362 (88%) | 22,250 (86%) | 24,032 (89%) | 22,453 (87%) |
| **Ratio of admissions between periods (wave 1 / baseline)[g]** | **Ratio** | **CI of ratio** | **Ratio[h]** | **CI of ratio** | **Ratio** | **CI of ratio** | **Ratio** | **CI of ratio** | **Ratio** | **CI of ratio** |
| All admissions | 0.83 | 0.82,0.84 | 0.73 | 0.72,0.74 | 0.84 | 0.82,0.85 | 0.87 | 0.86,0.89 | 0.97 | 0.95,0.98 |
| ICU admissions | 0.99 | 0.97,1.01 | 0.84 | 0.81,0.87 | 0.97 | 0.92,1.01 | 1.08 | 1.03,1.13 | 1.23 | 1.17,1.29 |
| Non-ICU admissions | 0.81 | 0.80,0.82 | 0.71 | 0.70,0.72 | 0.82 | 0.81,0.83 | 0.84 | 0.83,0.86 | 0.93 | 0.92,0.95 |

Notes

Abbreviation: NYC = New York City; CI = 95% confidence interval; ICU = intensive care unit

[a] Hospital quartiles are defined based on a hospital's 2017–2019 percentage of adult hospitalizations where the primary payer was Medicaid or self-pay. Hierarchical algorithm based on common payment logic was used to identify primary payer.

[b] Excludes specialty hospitals and hospitals without ICUs (N = 6).

[c] Pre-pandemic period is defined as CDC weeks 2–10 of 2020, corresponding to January 5-March 7, 2020 (based on date of admission).

[d] Wave 1 of COVID-19 is defined as CDC weeks 11–19 of 2020, corresponding to March 8-May 9, 2020 (based on date of admission).

[e] ICU use is defined based on revenue codes 200, 201, 202, 203,204, 207, 208, 209, 210, 211, 212, 213 and 219 (Weissman et al. 2017).

[f] Non-ICU use is defined as any admissions without revenue codes for ICU use but may include intermediate ICU use.

[g] The ratio of admissions in each quartile is statistically different from the ratios of admissions in all other quartiles at p < 0.05.

Source: New York State Department of Health, Statewide Planning and Research Cooperative System (SPARCS) 2020 inpatient data, 2022 release.

quartiles of hospitals. Compared with the baseline, the number of all-cause admissions city-wide declined by 17% in Wave 1. All-cause admissions by Q1-Q4 hospitals were 73%, 84%, 87%, and 97% of their baseline levels, respectively (all four ratios are statistically different from one another). The citywide number of ICU admissions remained virtually unchanged in Wave 1, but there was significant variation among hospitals across quartiles, with ICU admissions in Q1-Q4 hospitals being 84%, 97%, 108%, and 123% of the baseline levels, respectively (all four ratios are statistically different), consistent with the hypothesis that hospitals serving a higher proportion of Medicaid and uninsured patients pre-COVID experienced a greater surge of patients requiring critical care during the first COVID-19 wave.

Table 3 shows crude and risk-adjusted mortality rates in each period across the four hospital quartiles. The citywide number of in-hospital deaths among all admissions rose dramatically from 4,885 to 18,147, resulting in an increased crude mortality rate from 3.4 to 15.1 per 100 admissions. While both the crude and risk-adjusted mortality rates for all admissions were similar across quartiles pre-pandemic, there were large disparities in these rates in Wave 1. The crude mortality rates for all admissions in Wave 1 were 11.5 (95% confidence level (CI) = 11.1, 11.8), 16.4 (CI = 16.0,16.9), 18.9 (CI = 18.4,19.4), and 14.7 (CI = 14.2,15.1) per 100 admissions among Q1-Q4 hospitals, respective. After accounting for differences in patient-level characteristics, the risk-adjusted rates for all admissions were higher among Q3 and Q4 hospitals, 17.1 (CI = 16.6,17.6) and 16.8 (CI = 16.3,17.3) per 100 admissions, respectively, than Q1 and Q2 hospitals, 12.4 (CI = 12.0,12.8) and 14.9 (CI = 14.5,15.3) per 100 admissions,

**Table 3. Crude and risk-adjusted mortality rates among adults ages 18+ in NYC by hospital quartile, January 5-May 9, 2020[a].**

| | NYC | | Quartile 1 (lowest % of Medicaid or self-pay patients) | | Quartile 2 | | Quartile 3 | | Quartile 4 (highest % of Medicaid or self-pay patients) | |
|---|---|---|---|---|---|---|---|---|---|---|
| | (N = 47 hospitals)[b] | | (N = 11 hospitals) | | (N = 12 hospitals) | | (N = 12 hospitals) | | (N = 12 hospitals) | |
| | Pre-pandemic[c] | Wave 1[d] | Pre-pandemic | Wave 1 | Pre-pandemic | Wave 1 | Pre-pandemic | Wave 1 | Pre-pandemic | Wave 1 |
| **All in-hospital deaths** | 4,885 | 18,147 | 1,602 | 4,167 | 1,506 | 5,263 | 1,173 | 4,917 | 604 | 3,800 |
| Crude mortality rate per 100 [CI] | 3.4 [3.3–3.5] | 15.1 [14.9–15.3] | 3.2 [3.0–3.4] | **11.5** [11.1–11.8] | 3.9 [3.7–4.1] | **16.4** [16.0–16.9] | 3.9 [3.7–4.2] | **18.9** [18.4–19.4] | 2.2 [2.1–2.4] | **14.7** [14.2–15.1] |
| Risk-adjusted mortality rate per 100 [CI][e] | 3.4 [3.3–3.5] | 15.1 [14.9–15.3] | 3.2 [3.0–3.4] | **12.4** [12.0–12.8] | 3.5 [3.3–3.7] | **14.9** [14.5–15.3] | 3.7 [3.5–4.0] | 17.1 [16.6–17.6] | 3.3 [3.0–3.6] | 16.8 [16.3–17.3] |
| **ICU deaths[e]** | 2,261 | 6,778 | 743 | 1,729 | 560 | 1,628 | 565 | 1,740 | 393 | 1,681 |
| Crude mortality rate per 100 [CI] | 14 [13.5–14.6] | 42.5 [41.5–43.5] | 12.4 [11.5–13.3] | **34.5** [32.9–36.1] | 14.7 [13.5–16.0] | 44.2 [42.1–46.4] | 16.2 [14.9–17.6] | 46.2 [44.0–48.3] | 13.8 [12.5–15.2] | 48.3 [46.0–50.6] |
| Risk-adjusted mortality rate per 100 [CI] | 14.2 [13.6–14.9] | 42.5 [41.5–43.5] | 14 [13–15.1] | **36.4** [34.7–38.2] | 14 [12.8–15.3] | 43.6 [41.5–45.8] | 15 [13.7–16.3] | 45.9 [43.8–48.1] | 13.8 [12.4–15.4] | 45.7 [43.6–48.0] |
| **Non-ICU deaths[f]** | 2,624 | 11,369 | 859 | 2,438 | 946 | 3,635 | 608 | 3,177 | 211 | 2,119 |
| Crude mortality rate per 100 [CI] | 2.0 [2.0–2.1] | 10.9 [10.7–11.1] | 2.0 [1.8–2.1] | **7.8** [7.5–8.1] | 2.7 [2.6–2.9] | **12.8** [12.4–13.2] | 2.3 [2.1–2.5] | **14.3** [13.8–14.8] | 0.9 [0.8–1.0] | **9.4** [9.0–9.8] |
| Risk-adjusted mortality rate per 100 [CI] | 2.1 [2.0–2.2] | 10.9 [10.7–11.1] | 1.9 [1.7–2.0] | **8.6** [8.3–9.0] | 2.3 [2.1–2.4] | **10.9** [10.6–11.3] | 2.3 [2.1–2.5] | 12.6 [12.1–13.0] | 1.7 [1.5–2.0] | 12.1 [11.6–12.7] |

Notes

**Bolded number** indicates that the mortality rate for a quartile is statistically different from the rates for all other quartiles in Wave 1 ($p < 0.05$).

Abbreviation: NYC = New York City; CI = 95% confidence interval; ICU = intensive care unit

[a] Hospital quartiles are defined based on a hospital's 2017–2019 percentage of adult hospitalizations where the primary payer was Medicaid or self-pay. Hierarchical algorithm based on common payment logic was used to identify primary payer.

[b] Excludes specialty hospitals and hospitals without ICUs (N = 6).

[c] Pre-pandemic period is defined as CDC weeks 2–10 of 2020, corresponding to January 5-March 7, 2020 (based on date of admission).

[d] Wave 1 of COVID-19 is defined as CDC weeks 11–19 of 2020, corresponding to March 8-May 9, 2020 (based on date of admission).

[e] The following patient-level characteristics were used for risk adjustment: sex, age, race/ethnicity, primary payer, weighted Elixhauser comorbidity index, admission from nursing home, emergency admission, severity of illness, and principal diagnosis.

[f] ICU use is defined based on revenue codes 200, 201, 202, 203,204, 207, 208, 209, 210, 211, 212, 213 and 219 (Weissman et al. 2017).

[g] Non-ICU use is defined as any admissions without revenue codes for ICU use but may include intermediate ICU use.

Source: New York State Department of Health, Statewide Planning and Research Cooperative System (SPARCS) 2020 inpatient data, 2022 release.

respectively, consistent with the hypothesis that hospitals serving a higher proportion of Medicaid and uninsured patients pre-COVID experienced higher mortality rates during the first COVID-19 wave.

The city-wide crude rate of mortality for ICU admissions increased from 14 to 42.5 deaths per 100 admissions. Q1 hospitals had significantly lower crude and risk-adjusted mortality rates than hospitals in the other quartiles. Specifically, the risk-adjusted mortality rate for Q1 hospitals was 36.4 (CI = 34.7,38.2) per 100 ICU admissions vs. 43.6 (CI = 41.5,45.8), 45.9 (CI = 43.8,48.1), and 45.7 (CI = 43.6,48.0) per 100 ICU admissions for Q2-Q4 hospitals, respectively. As non-ICU admissions made up close to 90% of all admissions in both periods, crude and risk-adjusted rates for non-ICU admissions had similar patterns as the rates for all admissions.

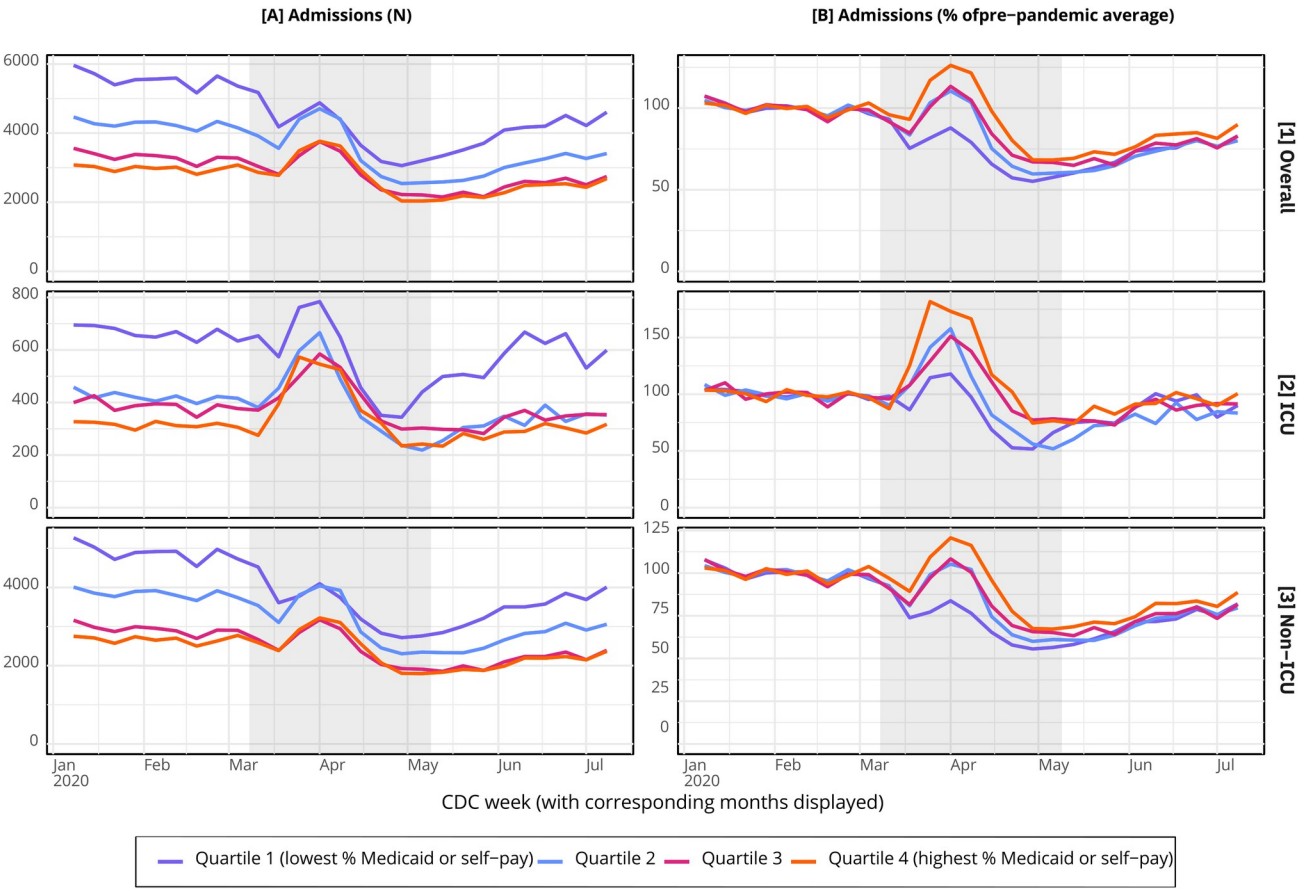

**Fig 1. Weekly hospital admissions among adults (18+) in New York City by hospital quartile, 2020.** Abbreviations: ICU = intensive care unit. *a* Hospital quartiles are defined based on a hospital's 2017–2019 percentage of adult hospitalizations where the primary payer was Medicaid or self–pay. Hierarchical algorithmbased on common payment logic was used to identify primary payer. Quartile 1 = lowest proportion of Medicaid or self-pay; Quartile 4 = highest proportion of Medicaid or self-pay. *b* Excludes specialty hospitals and hospitals without ICUs (N = 6). *c* Pre-pandemic period is defined as CDC weeks 2–10 of 2020, corresponding to January 5-March 7, 2020 (based on date of admission). *d* Wave 1 of COVID-19 is defined as CDC weeks 11–19 of 2020, corresponding to March 8-May 9, 2020 (based on date of admission). *e* ICU use is defined based on revenue codes 200, 201, 202, 203,204, 207, 208, 209, 210, 211, 212, 213 and 219 (Weissman et al. 2017). *f* Non-ICU use is defined as any admissions without revenue codes for ICU use but may include intermediate ICU use. Source: New York State Department of Health, Statewide Planning and Research Cooperative System (SPARCS) 2020 inpatient data, 2022 release.

As shown in S1-S6 Tables of S1 File, Hispanic and NH Black patients accounted for most of the increases in ICU demand and all three sets of deaths in the higher quartile hospitals.

Fig 1 shows that weekly ICU admissions to Q4 hospitals in late March were more than 175% of the baseline weekly average, substantially larger than the increase in ICU admissions experienced by Q1 hospitals. Fig 2 shows that both weekly ICU and non-ICU mortality reached their peaks in early April, approximately 800% and 2,500% of the baseline weekly averages among Q4 hospitals.

## Regression estimates

The full hierarchical logistic regression estimates for the three sets of inpatient deaths are presented in S7 Table of S1 File. The regression estimates show that being male and being older were risk factors for in-hospital mortality. Compared with NH Whites, Other race/ethnicity had a higher likelihood of ICU mortality while NH Blacks had a lower likelihood of non-ICU

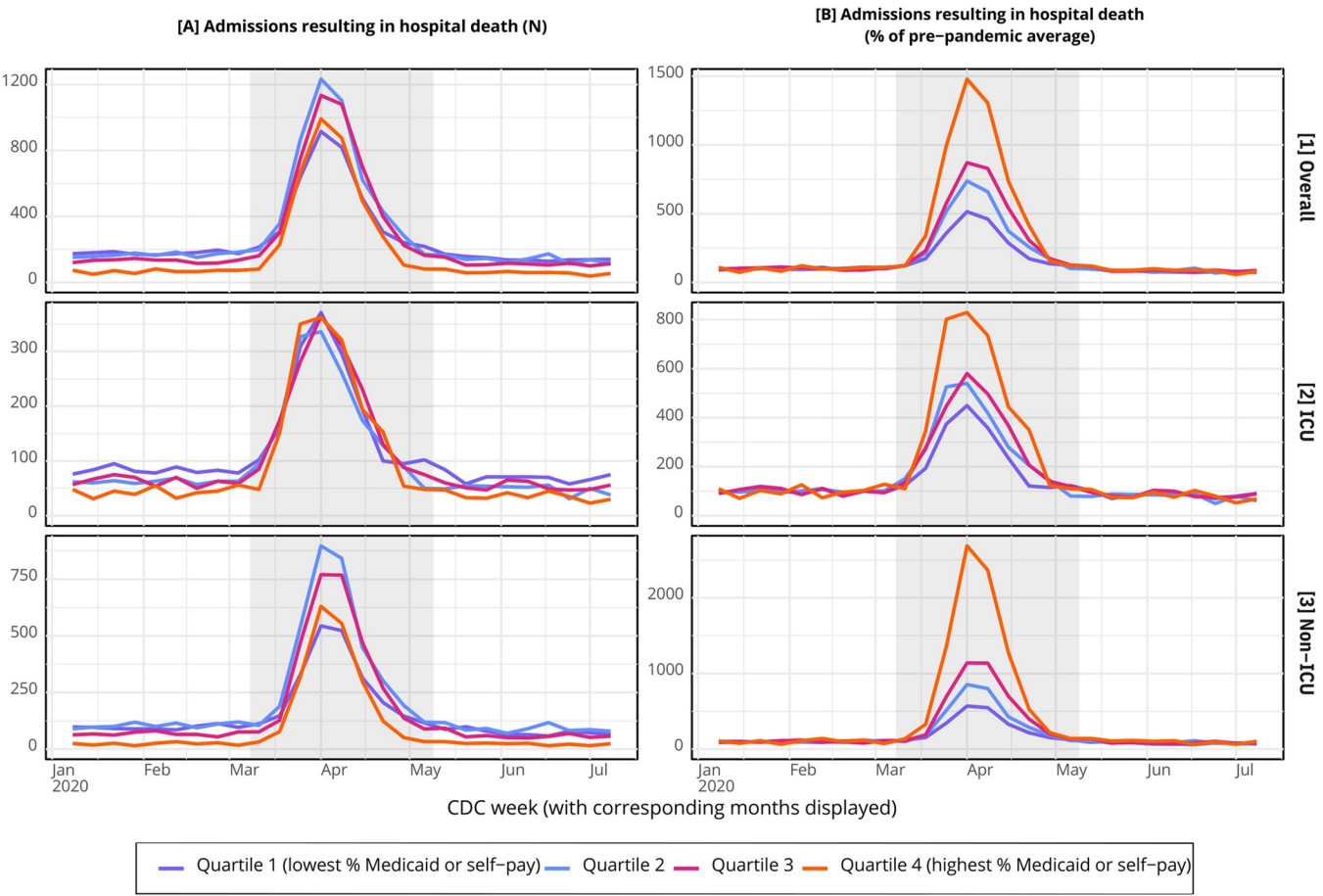

**Fig 2. Weekly inpatient mortality among adults (18+) in New York City by hospital quartile, 2020.** Abbreviations: ICU = intensive care unit. *a* Hospital quartiles are defined based on a hospital's 2017–2019 percentage of adult hospitalizations where the primary payer was Medicaid or self–pay. Hierarchical algorithmbased on common payment logic was used to identify primary payer. Quartile 1 = lowest proportion of Medicaid or self-pay; Quartile 4 = highest proportion of Medicaid or self-pay. *b* Excludes specialty hospitals and hospitals without ICUs (N = 6). *c* Pre-pandemic period is defined as CDC weeks 2–10 of 2020, corresponding to January 5-March 7, 2020 (based on date of admission). *d* Wave 1 of COVID-19 is defined as CDC weeks 11–19 of 2020, corresponding to March 8-May 9, 2020 (based on date of admission). *e* ICU use is defined based on revenue codes 200, 201, 202, 203,204, 207, 208, 209, 210, 211, 212, 213 and 219 (Weissman et al. 2017). *f* Non-ICU use is defined as any admissions without revenue codes for ICU use but may include intermediate ICU use. Source: New York State Department of Health, Statewide Planning and Research Cooperative System (SPARCS) 2020 inpatient data, 2022 release.

mortality. Compared with private insurance, the primary payers of Medicaid, Medicare, and Other were associated with a higher likelihood of mortality for both ICU and non-ICU admissions. ICU admissions from nursing homes were associated with a lower likelihood of mortality, but non-ICU admissions from nursing homes were associated with a higher likelihood of mortality. Having a higher Elixhauser comorbidity score and a major or extreme SOI were risk factors for both ICU and non-ICU mortality. Finally, compared with admissions to private AMCs and affiliates, those to non-AMC affiliated private hospitals (most of which are independent) were associated with a higher likelihood of ICU and non-ICU mortality.

Table 4 shows that critical care strain, the primary variable of interest, was associated with a higher likelihood of both ICU and non-ICU mortality. Compared with the reference group of 100% or less of the baseline weekly average, ICU admissions on a day for which the ICU volume was 101–150%, 151–200%, and > 200% of the baseline weekly average had odds ratios of 1.17 (CI = 1.10,1.26), 2.63 (CI = 2.31,3.00), and 3.26 (CI = 2.82,3.78) for inpatient mortality, and non-ICU admissions on a day for which the ICU volume was 101–150%, 151–200%,

**Table 4. Independent association between hospital strain and inpatient mortality among adults (18+) in New York City, January 11- May 9, 2020[a,b].**

| Outcome | ICU death[c] (N = 30,326 hospitalizations.) | | Non-ICU death[d] (N = 219,729 hospitalizations.) | | Overall inpatient death (N = 250,101 hospitalizations.) | |
|---|---|---|---|---|---|---|
| Hospital strain (ICU volume as % prepandemic baseline)[e] | Odds Ratio | 95% CI | Odds Ratio | 95% CI | Odds Ratio | 95% CI |
| 0% to 100% [Reference] | 1 | (1.00,1.00) | 1 | (1.00,1.00) | 1 | (1.00,1.00) |
| 101% to 150% | 1.17*** | (1.10,1.26) | 1.28*** | (1.22,1.34) | 1.28*** | (1.24,1.34) |
| 151% to 200% | 2.63*** | (2.31,3.00) | 2.60*** | (2.40,2.82) | 2.62*** | (2.45,2.80) |
| > 200% | 3.26*** | (2.82,3.78) | 3.44*** | (3.13,3.78) | 3.36*** | (3.11,3.63) |

Notes

* p<0.05

** p<0.01

*** p<0.001

Abbreviations: ICU = intensive care unit

[a] < 100 observations were excluded due to missing data.

[b] Estimates were generated from hierarchical logistical regressions, controlling for patient sex, age, race/ethnicity, primary payer, weighted Elixhauser comorbidity index, admission from nursing home, emergency admission, severity of illness (SOI), principal diagnosis, neighborhood of residence, and hospital type.

[c] ICU use is identified based on revenue codes 200, 201, 202, 203,204, 207, 208, 209, 210, 211, 212, 213 and 219 (Weissman et al. 2017).

[d] Non-ICU use is defined as any admissions without revenue codes for ICU use but may include intermediate ICU use.

[e] Hospital strain is defined as the total number of adult ICU admissions to the hospital in the week prior to and including the patient admission date as a percentage of the hospital's pre-pandemic baseline weekly average ICU admissions.

Source: New York State Department of Health, Statewide Planning and Research Cooperative System (SPARCS) 2020 inpatient data, 2022 release.

and > 200% of the baseline weekly average had odds ratios of 1.28 (CI = 1.22,1.34), 2.60 (CI = 2.40,2.82), and 3.44 (CI = 3.11,3.63) for inpatient mortality.

As shown in S8-S10 Tables of S1 File, the regression estimates for hospital strain were similar with alternative strain measures. Additionally, as reported in S11 Table of S1 File, the non-hierarchical logistic regression with hospital fixed effects produced similar estimates for the hospital strain and other variables.

## Discussion

The COVID-19 pandemic had a devastating and inequitable impact on NYC [31]. With three quarters of COVID-19 related deaths in NYC taking place in hospitals, which have been found to be highly segregated, this study examined the possible relationship between hospital segregation and the high rates of in-hospital mortality in NYC using all-payer hospital discharge data. We found that hospitals serving a high proportion of Medicaid or uninsured patients pre-COVID-19 pandemic experienced a greater strain in ICU capacity and larger crude and risk-adjusted mortality rates for both ICU and non-ICU admissions during the first wave of the pandemic. Additionally, we found that strained ICU capacity was independently associated with a greater likelihood of in-hospital death. Together, these findings suggest that a greater surge of patients needing critical care in segregated, under-resourced hospitals may have contributed to excess mortality rates, with Hispanic and NH Black patients accounting for most of the mortality increases.

Our study complements the existing literature in several ways. First, while strained critical care capacity has been linked to greater in-hospital mortality in both the pandemic and non-pandemic settings [18, 25, 26, 32], relatively few studies have assessed differences in critical care strain and outcomes across a census of hospitals within the same jurisdiction. Second, with few exceptions [25, 26], current studies on critical care strain have generally not examined the impact of critical care strain on non-ICU patients. Consistent with an analysis conducted

by Kadri et al. [26], we found that strained ICU capacity may be associated with worse outcomes among non-ICU patients, suggesting the importance of including both ICU and non-ICU outcomes in assessing the impact of critical care strain. Third, a recent study found that load imbalance was widespread in the US during the COVID-19 pandemic [33]. Additionally, the study found that hospitals serving high proportions of Medicaid patients and Black Medicare patients tended to be over capacity in regions experiencing load imbalance. Focusing on NYC, our study not only found similar results but further demonstrated the potential adverse impact of load imbalance and critical care strain on health outcomes. Finally, our study added to the accumulating body of evidence that segregation of care may lead to health disparities [14, 34].

Structural racism in historical and modern health care policy has contributed to people of color being disproportionately uninsured or covered by public insurance programs that offer lower reimbursement rates, a key driver of health care segregation [35]. Other racist policies, such as redlining, a federal program designed to residentially segregate Black and immigrant communities, also contributed to care segregation [36, 37]. While a comprehensive discussion of potential policy solutions for care segregation is beyond the scope of this paper, we offer the following recommendations. First, in order to incentivize providers to serve uninsured and Medicaid-insured patients, it is essential to institute payment reforms, such as raising Medicaid reimbursement rates and implementing hospital price regulation, and to expand health insurance coverage [38–41]. Second, it is important to enforce tax-exempt status for non-profit hospitals and ensure that benefits they provide for marginalized communities are commensurate to tax breaks and other government benefits received [42]. Finally, measures that advance anti-racism in clinical practice and care and policies that address broader social exclusion and residential segregation and promote reinvestment in historically redlined neighborhoods and minority service institutions are needed [35, 43, 44].

Our study findings may also inform preparedness and response in future pandemics and other health care emergencies. As all disease outbreaks are likely to have a disproportionate impact on marginalized populations [45], persistence of care segregation would lead to overburdened safety-net hospitals and worse outcomes in these facilities. Hence, it is crucial to strengthen support for the safety-net system and ensure its ability to meet unexpected increases in demand for ICU and other services, including addressing shortages of specialized health care providers [20]. Development of coordinated regional plans for load balancing is also necessary to promote equitable and timely access to critical care and prevent catastrophic outcomes among overburdened safety-net facilities during times of crisis, with an emphasis on supporting independent safety-net hospitals that may face additional hurdles in coordinating load balancing [46].

## Limitations

This study has several limitations. First, the discharge records included in SPARCS were provided by individual facilities, which may have inconsistent coding and reporting practices, resulting in issues such as misclassification of race/ethnicity that has been found to be more likely for Hispanic and Asian or Pacific Islander patients, potentially understating the impact of the pandemic on these groups in the analysis [47]. Second, our measures of hospital strain were imperfect proxies, as we did not have full information on each hospital's capacity to provide critical care and the demand for their critical care services. Third, the COVID-19 diagnosis code was not implemented until April 1, 2020; as a result, we could not identify many of the patients infected with COVID if they were discharged before that date. Nonetheless, as shown in S12 Table of S1 File, we obtained similar estimates for hospital strain if only admissions on

or after April 1 were included in the model. Fourth, patients who died in a hospital before they could be admitted were not included in the mortality analysis. Fifth, using the winter months of 2020, during which there may already be a high number of respiratory-related hospitalizations, as the baseline period could have underestimated the overall impact of COVID-19. Finally, while we accounted for many key variables, the estimates for hospital strain may still be biased by unobserved factors.

## Conclusions

Similar to many large cities in the US, NYC has a highly segregated healthcare system, with safety-net hospitals providing a disproportionate share of care for low-income and racial/ethnic minority patients. This population-level analysis of hospitalizations in NYC found that the greater strain in ICU capacity among safety-net hospitals was associated with higher likelihood of inpatient mortality during the first wave of the COVID-19 pandemic. Our findings are consistent with the hypothesis that health care segregation may have a substantial impact on health inequities and highlight the essential role of desegregating health care, strengthening support for the safety-net system, and reinvesting in redlined neighborhood during a pandemic and beyond in the promotion of health equity.

## Supporting information

**S1 File.**
(DOCX)

## Acknowledgments

We would like to acknowledge valuable feedback provided by participants in the Healthcare Segregation Workgroup at the New York City Department of Health and Mental Hygiene.

## Author Contributions

**Conceptualization:** Anna Zhilkova, Duncan Maru, Tsu-Yu Tsao, Michelle E. Morse.

**Data curation:** Anna Zhilkova, Laila Alsabahi.

**Formal analysis:** Anna Zhilkova, Laila Alsabahi, Tsu-Yu Tsao.

**Investigation:** Anna Zhilkova.

**Methodology:** Anna Zhilkova, Donald Olson, Tsu-Yu Tsao.

**Project administration:** Anna Zhilkova.

**Supervision:** Duncan Maru, Michelle E. Morse.

**Validation:** Laila Alsabahi.

**Visualization:** Anna Zhilkova, Laila Alsabahi.

**Writing – original draft:** Anna Zhilkova, Tsu-Yu Tsao.

**Writing – review & editing:** Anna Zhilkova, Laila Alsabahi, Donald Olson, Duncan Maru, Tsu-Yu Tsao, Michelle E. Morse.

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
