## [Decision Letter · Decision Letter 0]

2 Jan 2024

PONE-D-23-36641Hospital segregation, critical care strain, and inpatient mortality during the COVID-19 pandemic in New York CityPLOS ONE

Dear Dr. Tsao,

Thank you for submitting your manuscript to PLOS ONE. After careful consideration, we feel that it has merit but does not fully meet PLOS ONE’s publication criteria as it currently stands. Therefore, we invite you to submit a revised version of the manuscript that addresses the points raised during the review process.

Thank you for submitting this interesting manuscript.  We have provided reviewer critiques (I have acted as an academic reviewer for this manuscript with critiques provided below) which would help the readership of the journal interpret and understand this interesting data.  Please clarify whether there was an a priori hypothesis and the analysis plan.  Although relative differences in mortality from baseline is a reasonable way to present the data, it would be helpful to see absolute differences in mortality rates (without changes from the baselines) across the hospital quartiles and show whether there are major differences. 

We look forward to receiving your revised manuscript.

Kind regards,

Dong Wook Chang

Academic Editor

PLOS ONE

Journal Requirements:

3. We notice that your supplementary tables are included in the manuscript file. Please remove them and upload them with the file type 'Supporting Information'. Please ensure that each Supporting Information file has a legend listed in the manuscript after the references list.

Additional Editor Comments:

Academic Editor Review:

Thank you for submitting this interesting manuscript to PLOS One. I would like to raise the following critiques for your consideration.

This in an interesting manuscript that examines whether hospitals in NYC that serve a higher proportion of Medicaid and uninsured patients experienced greater strain and hospital mortality during the 1st wave of the COVID pandemic. Not surprisingly, the authors found that greater ICU strain was associated with higher mortality. They also present data to show that hospitals that serve more under-served patients (divided by quartiles) had higher mortality rates. Although the findings are interesting, there are several critiques which should be addressed:

Major issues:

1) Based on the description of the statistical analysis plan, it is not clear whether this is a descriptive analysis or whether there was an a priori hypothesis about the mortality across hospital quartiles. This needs to be specified. If there was a pre-specified hypothesis, the appropriate use of statistical tests that examine trends across the quartiles should be used and described.

2) For mortality (both ICU and non-ICU), the authors present relative changes compared to baseline (prior to pandemic). While this provides some interesting information, one can argue that the best outcome to see if there were truly differences in mortality across the hospital quartiles would be risk-adjusted mortality rates (not changes from baseline). These should also be reported in absolute values (i.e. mortality rate) rather than changes from baseline. COVID pneumonia was novel, so it is not clear that changes from a baseline (in which there was no COVID pneumonia) is as relevant as absolute mortality. When one looks at mortality trends during Wave 1 (Supplemental Table 2, Overall in-hospital deaths) across the hospital quartiles, it is not clear that there is an important trend or difference. The authors need to provide some clarity on these trends.

3) The figures are very difficult to read (resolution) so it was challenging to understand the results. The supplementary tables are helpful, but there is an overwhelming amount of data. The authors should improve the clarity and quality of the figures (more informative figure legends would also be helpful) or provide tables that show key results in a way that readers could easily interpret.

Minor Issues:

1) Please provide a brief justification for why Wave 1 was used and not the entire duration of the COVID pandemic (or include additional waves)

2) In the statistical analysis section (page 7, lines 152-153), the authors state that “Possible changes in the composition of admissions between the two periods were also examined.” Please clarify what was examined and how.

3) Please provide some more details on how patient acuity and severity of illness are captured in the database (Page 7, lines 161-162)

4) It might be more appropriate to discuss how this work fits into the existing body of literature rather than offering recommendations to reduce care segregation (Discussion, Page 10-11). The paragraph on how this data might inform preparedness and response in future pandemics is excellent.

Reviewers' comments:

Reviewer's Responses to Questions

**Comments to the Author**

1. Is the manuscript technically sound, and do the data support the conclusions?

Reviewer #1: Yes

2. Has the statistical analysis been performed appropriately and rigorously? 

Reviewer #1: Yes

3. Have the authors made all data underlying the findings in their manuscript fully available?

Reviewer #1: Yes

4. Is the manuscript presented in an intelligible fashion and written in standard English?

Reviewer #1: Yes

5. Review Comments to the Author

Reviewer #1: The current study describes an important and robust analysis of hospitalization and mortality trends in NYC hospitals during the height of the COVID-pandemic. This analysis is innovative and clearly portrays disparities in hospital burden and negative outcomes associated with safety net hospitals that serve the majority of poor and un or under insured patients. I commend the authors on this thoughtful and important study. I only have some minor suggestions for some clarifications as follows:

1. It may be helpful to explicitly introduce the H+H system in NYC, which is the primary public and safety-net hospital in NYC. This may not be typical of other cities or well-known to non NYC readers, even though many of the highest risk hospitals in the study were part of the system.

2. It is not immediately clear how to interpret patients with "self-pay" - in these data, would this be indicative of being uninsured or under-insured? Technically some very wealthy patients may "self-pay" as well, so would be helpful to explain to the reader how to interpret this and why authors decided to group this type of payment with Medicaid when creating the quartiles.

3. Authors note in the methods "Several alternative proxies for hospital strain were used to assess sensitivity of regression estimates to hospital strain measures" while the results are included in supplementary tables, it would be helpful if authors explicitly listed what was tested.

4. In the results, it would be helpful to start with the total number of hospitals that were examined and how many were distributed across quartiles, to get a sense of spread and volume of hospitals.

5. Authors say "patients of color" when describing some of the results, but they never indicate what race/ethnicity groups they are included in this definition.

6. On line 206 authors say most regression estimates were consistent with previous studies - what previous studies is this referring to? Please reference appropriately

7. All the figures are blurry and I am not able to see or read the labels - please produce higher quality images

8. The figures and tables hold an overwhelming amount of information - I may suggest presenting findings only by non-ICU and ICU and remove the third "all hospitalization" categories, to reduce the amount of information being presented and interpreted.

6. PLOS authors have the option to publish the peer review history of their article (what does this mean?). If published, this will include your full peer review and any attached files.

Reviewer #1: No

---

## [Author Response · Author response to Decision Letter 0]

1 Mar 2024

The Response to Reviewers is attached as a document.

---

## [Editor Report · Decision Letter 1]

18 Mar 2024

Hospital segregation, critical care strain, and inpatient mortality during the COVID-19 pandemic in New York City

PONE-D-23-36641R1

Dear Dr. Tsao,

We’re pleased to inform you that your manuscript has been judged scientifically suitable for publication and will be formally accepted for publication once it meets all outstanding technical requirements.

Kind regards,

Dong Wook Chang

Academic Editor

PLOS ONE

Additional Editor/Reviewer Comments (optional):

The authors have addressed reviewers' critiques sufficiently. The revisions have significantly strengthened the quality of the manuscript.

---

## [Editor Report · Acceptance letter]

1 Apr 2024

PONE-D-23-36641R1 

PLOS ONE

Dear Dr. Tsao, 

I'm pleased to inform you that your manuscript has been deemed suitable for publication in PLOS ONE. Congratulations! Your manuscript is now being handed over to our production team.

Kind regards, 

on behalf of

Dr. Dong Wook Chang 

Academic Editor

PLOS ONE